# PHASE TRANSITIONS IN THE OUTPUT DISTRIBUTION OF LARGE LANGUAGE MODELS

## ABSTRACT

In a physical system, changing parameters such as temperature can induce a phase transition: an abrupt change from one state of matter to another. Analogous phenomena have recently been observed in large language models. Typically, the task of identifying phase transitions requires human analysis and some prior understanding of the system to narrow down which low-dimensional properties to monitor and analyze. Statistical methods for the automated detection of phase transitions from data have recently been proposed within the physics community. These methods are largely system agnostic and, as shown here, can be adapted to study the behavior of large language models. In particular, we quantify distributional changes in the generated output via statistical distances, which can be efficiently estimated with access to the probability distribution over next-tokens. This versatile approach is capable of discovering new phases of behavior and unexplored transitions – an ability that is particularly exciting in light of the rapid development of language models and their emergent capabilities.

## 1 INTRODUCTION

In physics, a phase transition refers to an abrupt change in a system's macroscopic behavior due to variations in external conditions such as temperature or pressure. A familiar example is the transition of water from a phase of high density (liquid) to one of low density (gas) at the boiling point. Phenomena reminiscent of such classical phase transitions have also been observed in neural networks (NNs), including rapid knowledge acquisition and sudden improvements in capabilities (Shwartz-Ziv & Tishby, 2017; McGrath et al., 2022; Olsson et al., 2022). For instance, during training, AlphaZero exhibited periods where increasingly sophisticated chess openings were suddenly favored (McGrath et al., 2022), and large language models (LLMs) have shown sudden improvements linked to the formation of specialized circuitry known as induction heads (Olsson et al., 2022).

As these phase transitions in deep learning systems often coincide with points where model capabilities emerge, disappear, or simply fundamentally change, their systematic detection promises to enable a more streamlined interrogation of the learning process, potentially leading to better training methods or safer model deployment. (Ganguli et al., 2022; Liu et al., 2022). For example, analyzing the grokking phenomenon led to strategies for accelerating generalization (Liu et al., 2022). Existing methods often require significant human input and may miss critical transitions (Zhong et al., 2023), highlighting the need for automated detection techniques.

Inspired by statistical methods in physics that require minimal prior knowledge (Carrasquilla & Melko, 2017; Van Nieuwenburg et al., 2017), we introduce an automated approach for detecting phase transitions in LLMs using statistical distances from the family of $f$-divergences. This method provides a versatile tool for objectively mapping out the phase diagrams of generative models. Such an approach has the potential to characterize unexplored transitions and discover new behavioral phases.

As a demonstration, we characterize transitions occurring in Pythia (Biderman et al., 2023), Mistral (7B) (Jiang et al., 2023), and Llama 3 (8B) (AI@Meta) language models as a function of three control parameters: an integer in the input prompt, the temperature hyperparameter for text generation, and the model's training epoch. We obtain the following findings:

- Varying input: Our analysis identifies sharp transitions in text output linked to integer tokenization and reveals that instruction-tuned models like Llama 3 and Mistral can order integers, whereas the base models can not.

- Varying temperature: We map out distinct behavioral phases as a function of temperature — specifically, a deterministic "frozen" phase at low temperatures, an intermediate "coherent" phase, and a "disordered" phase at high temperatures. The behavior as a function of system size for the higher temperature transition shows signs of critical behavior. We observe that LLMs can exhibit a negative "heat capacity", where its mean energy decreases as temperature increases.

- Varying training epoch: We also find that rapid changes in weight distributions during training correspond to transitions in output behavior. The exact location of the output transitions can be prompt-dependent.

## 2 METHODOLOGY

### 2.1 QUANTIFYING DISSIMILARITY BETWEEN DISTRIBUTIONS

In this work, we view phase transitions as rapid changes in the probability distribution $P(\cdot|T)$ governing the state of a system $\boldsymbol{x} \sim P(\cdot|T)$ as its control parameter $T$ is varied.[1] That is, values of the parameter at which the distribution changes strongly are considered critical points where phase transitions occur. While it is possible to generalize our approach to distributions conditioned on multiple control parameters as in (Arnold et al., 2023b;a), in the following we consider the one-dimensional scenario for simplicity.

We quantify the rate of change using $f$-divergences (Liese & Vajda, 2006), as they have particularly nice properties, such as satisfying the data processing inequality. Given a convex function $f : \mathbb{R}_{\geq 0} \to \mathbb{R}$ with $f(1) = 0$, the corresponding $f$-divergence is a statistical distance defined as

$$D_f[p, q] = \sum_{\boldsymbol{x}} q(\boldsymbol{x}) f\left(\frac{p(\boldsymbol{x})}{q(\boldsymbol{x})}\right) \geq 0. \tag{1}$$

Prominent examples of $f$-divergences include the Kullback-Leibler (KL) divergence, the Jensen-Shannon (JS) divergence, as well as the total variation (TV) distance.

The TV distance and the JS divergence have also had tremendous success in detecting phase transitions in physical systems without prior system knowledge under the name of "learning-by-confusion" (Van Nieuwenburg et al., 2017).[2]

### 2.2 DETECTING PHASE TRANSITIONS

Having defined appropriate notions of distance between probability distributions, we now describe their use to detect phase transitions: Consider a uniform one-dimensional grid $\mathcal{T}$ of control parameter values $T$. For each midpoint $T^*$ lying halfway in between grid points, we define a segment of grid points $\sigma_{\text{left}}(T^*)$ to the left and a segment of grid points $\sigma_{\text{right}}(T^*)$ to the right of $T^*$. Denoting the cardinality as $|\cdot|$, we define

$$P(\boldsymbol{x}|i, T^*) = \frac{1}{|\sigma_i(T^*)|} \sum_{T \in \sigma_i(T^*)} P(\boldsymbol{x}|T) \tag{2}$$

for both $i \in \{\text{left}, \text{right}\}$. Intuitively, $P(\boldsymbol{x}|i, T^*)$ encodes the probability that the system is in state $\boldsymbol{x}$ conditioned on the fact that the temperature is uniformly distributed over the temperatures contained

---

[1]This definition encompasses phase transitions in physics, i.e., abrupt changes in the distribution governing large-scale systems of interacting constituents.

[2]Note that both the TV distance and the JS divergence form lower bounds to the KL divergence and other $f$-divergence, such as the $\chi^2$ divergence: $D_{\text{JS}}[p, q] \leq D_{\text{TV}}[p, q] \leq \sqrt{D_{\text{KL}}[p, q]} \leq \sqrt{D_{\chi^2}[p, q]}$ (Flammia & O'Donnell, 2023). In this sense, detecting a large dissimilarity in terms of the TV distance or the JS divergence also signals a large dissimilarity in other measures.

in $\sigma_i(T^*)$. Critical points in $T^*$ where phase transitions occur can then be identified as local maxima in the dissimilarity $D(T^*) = D\left[P(\cdot|\text{left}, T^*), P(\cdot|\text{right}, T^*)\right]$.

For the sake of simplicity, we proceed with segments of equal length in this article, and define the length $L = |\sigma_{\text{left}}| = |\sigma_{\text{right}}|$ as the number of parameter values $T$ to the left or right of $T^*$ that characterize the segment. We are free to adjust it according to the problem, as $L$ sets a natural length scale on which changes in the distributions are assessed. Different settings of $L$ will be discussed in Sec. 3.

## 2.3 Numerical Implementation for Language Models

In the case of language models, $\boldsymbol{x}$ is the sampled text and $T$ is any variable that influences the sampling probability. For tractable density, as is the case for autoregressive language models, we can efficiently sample text $\boldsymbol{x}$ for a given prompt and directly evaluate its probability $P(\boldsymbol{x}|T)$. Thus, we can obtain an unbiased estimate $\hat{D}(T^*)$ of $D(T^*)$ by replacing expected values in Eq. (1) with sample means. The samples correspond to text generated with language models conditioned on different parameter settings $T$, see Appendix A for details on implementation.

To increase the sampling efficiency and numerical stability, we rewrite our dissimilarity measures in a slightly different form. To this end, when sampling around $T^*$, we use Bayes' theorem on Eq. (2) to write the probability for a sample $\boldsymbol{x}$ to stem from segment $\sigma_i(T^*)$ as

$$P(\sigma_i|\boldsymbol{x}, T^*) = \frac{P(\boldsymbol{x}|i, T^*)}{P(\boldsymbol{x}|\text{left}, T^*) + P(\boldsymbol{x}|\text{right}, T^*)}. \tag{3}$$

This quantity is bounded between 0 and 1 and therefore more stable than the ratio of probabilities, which is the argument of $f$ in the $f$-divergence [Eq. (1)]. Specifically, using $P(\sigma_i|\boldsymbol{x}, T^*)$ as argument of a function $g$, we introduce the new quantity

$$D_g(T^*) = \frac{1}{2L} \sum_{i \in \{\text{left}, \text{right}\}} \sum_{T \in \sigma_i} \mathbb{E}_{\boldsymbol{x} \sim P(\cdot|T)} \left[ g\left[P(\sigma_i|\boldsymbol{x}, T^*)\right]\right], \tag{4}$$

which we will refer to as $g$-dissimilarity. Algorithm 1 provides pseudo-code for the sampling procedure with $L = 1$ and the $T$ parameter corresponding to model temperature.

---

**Algorithm 1** For a given language model implemented in code as "model" and a $g$-function implemented as "g_function", the $g$-dissimilarity defined in Equation (4) can be estimated at $T^* = \frac{1}{2}(T_n + T_{n+1})$ for $L = 1$ as:

**Input:** model, $n$, $T_n$, $T_{n+1}$, $n_{\text{samples}}$, g_function
**Output:** g-dissimilarity $D_g[T^*]$

1: $D_g = 0$
2: **for** $i$ in $\{1, \ldots, n_{\text{samples}}\}$ **do**
3:     **for** $j$ in $\{n, n+1\}$ **do**
4:         sample = model.generate(temperature=$T_j$)
5:         $p_n$ = model.evaluate_probability(sample, temperature=$T_n$)
6:         $p_{n+1}$ = model.evaluate_probability(sample, temperature=$T_{n+1}$)
7:         $D_g = D_g + $ g_function($p_j/(p_n + p_{n+1})$)
8:     **end for**
9: **end for**
10: $D_g = D_g/(2n_{\text{samples}})$

---

## 2.4 Discussion:
### $f$-$g$ correspondence, choice of $g$, connection to the Fisher Information

The $g$-dissimilarities [Eq. (4)] and the $f$-divergences [Eq. (1)] defined above correspond to each other in the following sense: any $g$-dissimilarity $D_g$ can be rewritten in the form of an $f$-divergence $D_f$ with

$$f(x) = \frac{x}{2} \cdot g\left(\frac{x}{1+x}\right) + \frac{1}{2} \cdot g\left(\frac{1}{1+x}\right), \tag{5}$$

see Appendix B for the derivation and further discussion. In particular, for the choice $g(x) = \log(x) + \log(2)$, $D_g$ corresponds to the JS divergence. For $g(x) = 1 - 2\min\{x, 1 - x\}$, $D_g$ corresponds to the TV distance.

A natural choice for $g$ is any linear function in $x$. In particular, setting $g(x) = 2x - 1$ results in a dissimilarity measure that quantifies the ability of an optimal classifier to tell whether a sample $x$ has been drawn in the left or right sector. This measure is 0 if the two distributions are completely indistinguishable and 1 if the two distributions are perfectly distinguishable. Moreover, $g(x) = 2x - 1$ is bounded between 1 and -1, where the edge values are attained for the certain predictions of 0 and 1, and the value 0 corresponds to uncertain predictions at 0.5. This results in a low variance and favorable convergence properties for $\hat{D}_{g(x)=2x-1}$, which we will refer to as *linear dissimilarity* in what follows.

This quantity is a valid $f$-divergence and, as shown in Appendix B, reduces to the Fisher information $\mathcal{F}$ (Amari & Cichocki, 2010) in lowest non-vanishing order of the distance $\delta T$ between neighboring parameter points on the grid. In fact, any $g$-dissimilarity with $g(1/2) = 0$ and a twice-differentiable $g$-function can be shown to be proportional to the Fisher information in lowest order. To give a sketch of the derivation, for $L = 1$ we can write $D_f(T^*) = \frac{1}{2}f''(1)\mathcal{F}(T^*)\delta T^2 + \mathcal{O}(\delta T^3)$.

Having the Fisher information as a limiting case is a desirable property: It is a well-known, generic statistical measure for quantifying how sensitive parameterized probability distributions are to changes in their parameters and its behavior is well-understood when used to detect phase transitions in physical systems (You et al., 2007; Gu, 2010; Prokopenko et al., 2011; Arnold et al., 2023a). When considering parametric distributions underlying physical systems, the Fisher information can be shown to be directly related to important physical quantities, such as the heat capacity and magnetic susceptibility (Arnold et al., 2023a). Physicists routinely detect phase transitions in physical systems by identifying divergences (in the case of infinite-sized systems) or sharp local maxima (in the case of finite-sized systems) in these quantities. While they have no direct correspondence in the more abstract realm of language models, our method extends this methodology to this domain allowing for the detection of analogous "phase transitions". The Fisher information provides a solid theoretical basis for applying concepts from statistical physics to LLMs, bridging the gap between physical systems and computational models.

Appendix D introduces physics-based concepts used in this paragraph and article, which may be unfamiliar to non-physicist readers.

## 2.5 Utilized Large Language Models

In this work, we study transitions in models of the Pythia, Mistral, and Llama families.

**Pythia** Pythia (Biderman et al., 2023) is a suite of 16 LLMs released in 2023 that were trained on public data in the same reproducible manner ranging from 70 million (M) to 12 billion (B) parameters in size. We consider every second model, i.e., the models with 70M, 410M, 1.4B, and 6.9B parameters.

**Mistral** From the Mistral family, we consider the base model Mistral-7B-v0.1 with 7.3B parameters and the corresponding fine-tuned Mistral-7B-Instruct model (Jiang et al., 2023) released in 2023.

**Llama** Llama 3 (AI@Meta) from Meta AI was released in 2024. We consider both the Llama-3 8B parameter base model and NVIDIA's chat-tuned Llama3-ChatQA-1.5-8B (Liu et al., 2024). For the chat model we use accordingly formatted inputs.

## 3 Results

In the following, we will explore all three fundamental ways in which a parameter $T$ may influence the output distribution of a language model: $(i)$ As a variable within the input prompt, we scan through integers injected to the prompt in Sec. 3.1. $(ii)$ As a hyperparameter controlling how a trained language model is applied, we vary the temperature in Sec. 3.2. $(iii)$ As a training hyperparameter of the language model, we vary the number of training epochs in Sec. 3.3.

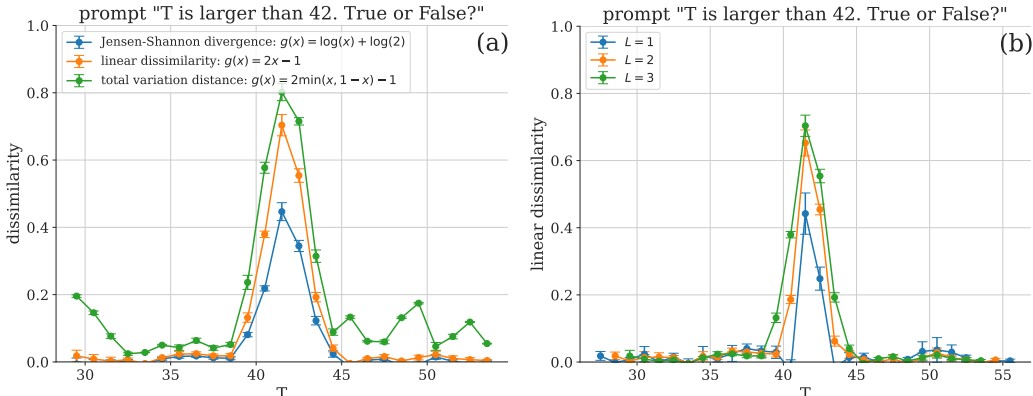

Figure 1: Mistral model applied to the integer ordering prompt. (a) Different $g$-dissimilarities with $L = 3$. (b) Linear dissimilarity for different $L$-values. [Number of text outputs generated per parameter value $T$: 10280. Number of generated output tokens: 10. Error bars indicate standard error of the mean over 4 batches, each with batch size 2056.] Appendix Fig. E1 provides additional scans for numbers different from $42$ and different formulations of the prompt, showcasing the robustness of these results under variations.

### 3.1 TRANSITIONS AS A FUNCTION OF A VARIABLE IN THE PROMPT

As an introduction, we start with the simplest case: The parameter $T$ to be varied is a particular part of the prompt, and all parameters of the language model itself are fixed. As a first such prompt, consider *"T is larger than 42. True or False?"* with an integer $T$ as the control parameter. An LLM that understands the order of integers should output very different answers for $T < 42$ versus $T > 42$, i.e., its distribution over outputs should change drastically around $T = 42$. Thus, in such a case we expect the dissimilarities to show a clear peak around $T = 42$.

Figure 1(a) shows dissimilarities based on various $g$-functions for the Mistral-7B-Instruct model (Jiang et al., 2023). All dissimilarities show a clear peak around $T = 42$, whereas they are relatively flat otherwise. This is a clear example of an abrupt transition between two distinct phases of behaviors of an LLM as a function of a tunable parameter. As compared to the linear dissimilarity, the logarithm-based JS divergence is arguably a bit sharper in that it decays more rapidly to baseline 0. The TV distance's peak is the broadest due to the $\min$ function appearing in its $g$-function. In the following, we will focus on the linear dissimilarity as a compromise between sensitivity and numerical stability.

The transition is also clearly visible using different $L$ settings, see Fig. 1(b). Smaller $L$ values are closer to the Fisher information limit, while larger values generally lead to higher distinguishability of distributions and therefore larger peaks at transition points. As we will see in more detail in Sec. 3.3, they can also be less susceptible to outliers due to the averaging over several parameter points.

Interestingly, when performing the same analysis on base models such as the Llama3-8B and Mistral base models, as well as Pythia models (Biderman et al., 2023) of various sizes, the resulting linear dissimilarity is flat, signaling the absence of any transition [see Fig. 2(a)]. In contrast to Mistral-7B-Instruct and NVIDIA's chat-tuned Llama3-8B, these models do not show a clear peak around $T = 42$.

A transition of a different origin can be observed in Fig. 2(b), where the LLMs are probed using the prompt *"T"* with $T$ again being an integer. Interestingly, all Pythia models show a peak between $T = 2020$ and $T = 2021$. This behavioral transition may originate from a transition in the tokenizers of these models, which encode numbers in a range below $T = 2021$ with a single token and numbers in a range at and above $T = 2021$ with two. This explanation is corroborated by the absence of the transition around $T = 2021$ for the Llama and Mistral models, whose tokenizers translate a number into tokens following rules that are independent of the number's frequency.

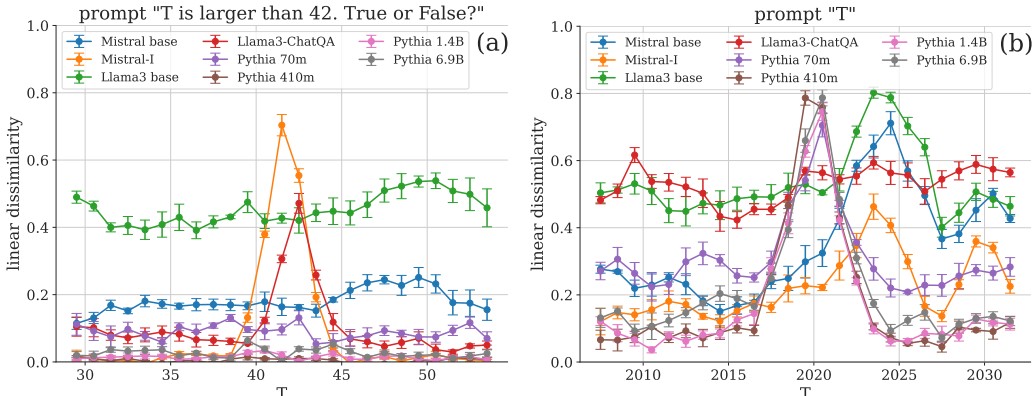

Figure 2: Benchmarking of various models using the linear dissimilarity with $L = 3$. (a) Test of ability to compare integers in value. (b) Bare integers as prompt reveals transition in tokenizer encoding. [Same numerical settings as in Fig 1.]

The Mistral models and the base Llama3-8B model show a smaller peak around $T = 2023/2024$. Both models have only encountered training data from before and around that time given their release date in 2023/2024, which may explain the peak. This transition is absent in the Pythia models.

## 3.2 TRANSITIONS AS A FUNCTION OF THE MODEL'S TEMPERATURE

Next, we consider transitions as a function of the temperature hyperparameter $T$ controlling how the logits $\boldsymbol{z}$ are converted to probabilities

$$p_i = \frac{e^{z_i/T}}{\sum_j e^{z_j/T}} \tag{6}$$

for next-token prediction. The sum runs over all possible tokens. Per construction, at $T = 1$ language models predict probabilities $p_i$ to approximate the distribution to be learned. In the limit $T \to 0$, the model deterministically picks the most likely next token in each step. For $T \to \infty$ the model samples the next token uniformly.

This scenario somewhat resembles a one-dimensional lattice of spins that are coupled via long-range interactions, i.e., the one-dimensional Ising model (Dyson, 1969; Martínez-Herrera et al., 2022), which has an order-disorder phase transition. In our case, the tokens take the role of the spins, and the coupling is mediated via the transformer's attention mechanism. Appendix D.3 elaborates on this analogy to Ising-type models.

The behavior observed in Fig. 3 (a) shares some interesting similarities with critical phenomena observed in physical systems. As the system size (output length) increases from 10 to 1000 tokens, we see a peak in dissimilarity becoming sharper and higher around $T = 0.75$. This sharpening of the peak with increasing system size is reminiscent of finite-size scaling effects seen near critical points in statistical physics, such as in the Ising model.

For comparison and to confirm the observed behavior, in Fig. 3 (b) we present a second analysis, independent of the dissimilarity-based indicators and taking inspiration from statistical mechanics, where the state of thermal systems is governed by the Boltzmann distribution (see Appendix C for details). We view the LLM as such a thermal system at varying temperature where the negative logarithmic probability at $T = 1$, $-\log P(\boldsymbol{x}|T = 1)$, takes on the role of the energy $E$ of a given text output $\boldsymbol{x}$. In physical systems governed by Boltzmann distributions, thermal phase transitions can be detected as peaks in the heat capacity $C(T) = \partial \mathbb{E}_{\boldsymbol{x} \sim P(\cdot|T)} [E(\boldsymbol{x})] / \partial T$, i.e., by looking at the temperature derivative of the mean total energy (Blundell & Blundell, 2009).

In the LLM case, the text outputs are not truly sampled from a Boltzmann distribution governed by the total energy. Instead, each individual token is drawn from a Boltzmann distribution for its individual energy conditioned on the previous tokens only. This procedure corresponds to a greedy

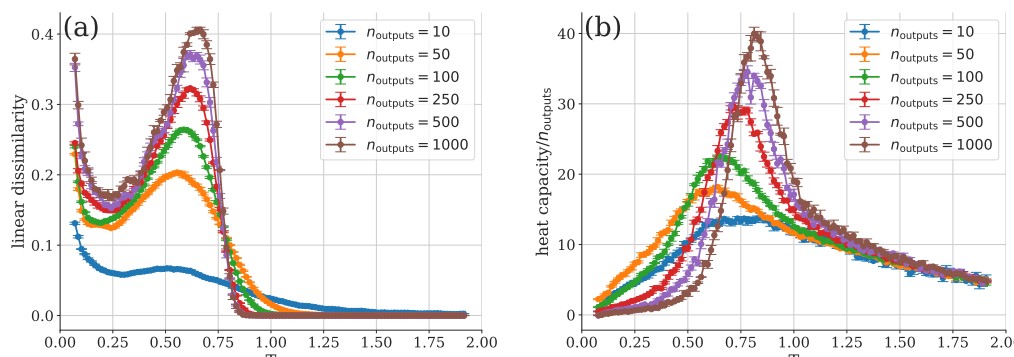

Figure 3: High-temperature transition of Pythia 70M model in response to the prompt "There's measuring the drapes, and then there's measuring the drapes on a house you haven't bought, a" – an excerpt from OpenWebText (Aaron Gokaslan and Vanya Cohen, 2019). (a) Linear dissimilarity measure ($L = 5$) and (b) heat capacity for various numbers of generated output tokens $n_{outputs}$ with the temperature range $[10^{-4}, 2]$. [Number of text outputs generated per parameter value $T$: 5000, 5000, 5000, 1500, 500, and 300 for $n_{outputs} = 10, 50, 100, 250, 500$, and 1000, respectively. Error bars indicate standard error of the mean over 5, 5, 5, 5, 5, and 3 batches for $n_{outputs} = 10, 50, 100, 250, 500$, and 1000, respectively.] Additional scans of the temperature transitions can be found in Appendix E.

sampling strategy. The resulting sampling mismatch can lead to the counterintuitive phenomenon of the mean energy of the system increasing with decreasing temperature corresponding to a negative "heat capacity", cf. Fig. 4 (b).

Zooming in on the system with only 10 output tokens, Fig 4 shows again the transition between sensible output and random output in panel (a), and a second transition at very low temperature in panel (b), where the model "unfreezes" from producing a deterministic output to producing sensible and varied text.

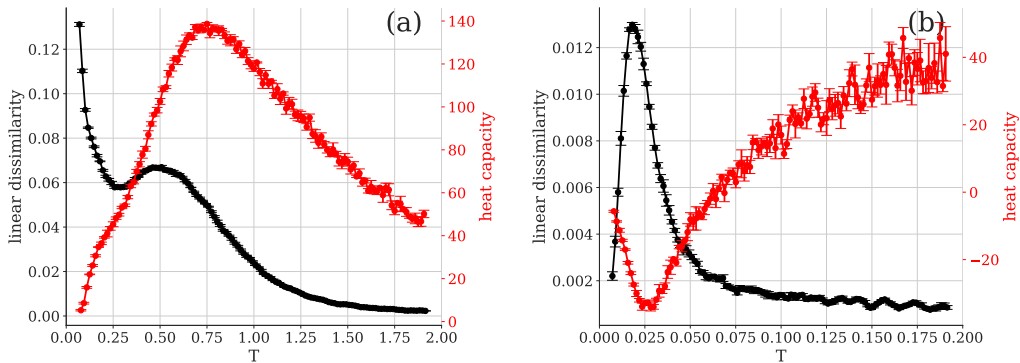

Figure 4: Temperature transitions of Pythia 70M model in response to the prompt "There's measuring the drapes, and then there's measuring the drapes on a house you haven't bought, a" – an excerpt from OpenWebText Aaron Gokaslan and Vanya Cohen (2019). Linear dissimilarity measure ($L = 5$) is shown in black. Heat capacity is shown in red. Dashed lines indicate local maxima, i.e., predicted critical points. Shaded regions indicate the error bands. (a) Temperature range $[10^{-4}, 2]$. (b) Zoomed-in range $[10^{-4}, 0.2]$ near $T = 0$. [Number of text outputs generated per parameter value $T$: 20480. Number of generated output tokens: 10. Error bars indicate standard error of the mean over 4 batches, each with batch size 5120.]

In Figs. 3 and 4 we have investigated the output distributions corresponding to a specific prompt. While we find that the temperature behavior is strongly dependent on the prompt, there seems to be a

trend: many distinct prompts lead to a transition at $T \approx 1$ (i.e., on the order of the natural temperature scale), at $T \ll 1$, or both. Further scans are shown in Appendix F.

The transition at low temperatures has recently been investigated by Bahamondes for GPT-2 using physics-inspired quantities. Moreover, they speculated on the existence of a phase transition at higher temperatures. In parallel to our work presented here, this higher temperature transition was analyzed by (Nakaishi et al., 2024) using correlations in part-of-speech (POS) tags. Their analysis revealed signs of critical behavior complementary to the peak behavior from Fig. 3.

### 3.3 TRANSITIONS AS A FUNCTION OF THE TRAINING EPOCH

Finally, we search for transitions as a function of the training epoch, i.e., we compare the output distributions of models at different stages during training and see whether there are certain epochs at which these statistics change drastically. This temporal analysis is the most challenging example, as the distributional changes are very noisy due to the inherently noisy training process of methods from the stochastic gradient descent family. Here, we analyze the Pythia suite of models for which such checkpoints are publicly available.

Millidge analyzed the weight distribution of the Pythia models, and similar weight-based analyses of other NNs during training have also been performed in previous works (Shwartz-Ziv & Tishby, 2017; Achille et al., 2017; Chen et al., 2023). In order to study the previously observed transitions (Millidge), we analyze changes in the weight distributions in the same manner as for the output distributions (see Sec. 2), i.e., to characterize phase transitions using dissimilarities. The lists of model weights are converted to distributions via histogram binning (10000 bins for the range -3 to 3).

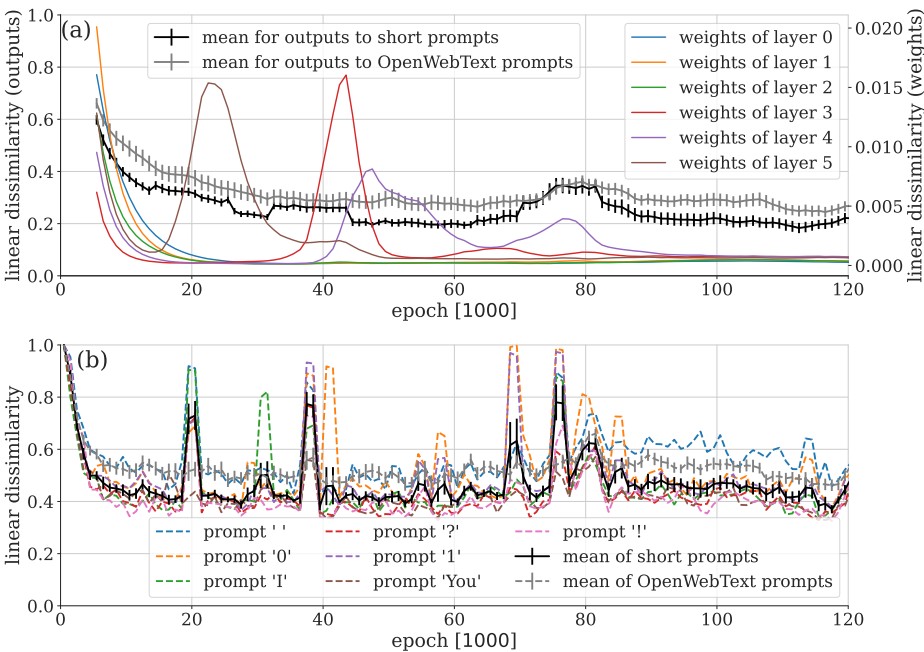

Figure 5: Linear dissimilarity by epoch, with checkpoints taken every 1000 epochs. (a) Computed at $L = 6$ for both weights and responses to 20 random prompts from OpenWebText (gray) and 7 short prompts (black) shown in panel (b). (b) Computed at $L = 1$ for several prompts. For reference, the mean linear dissimilarity over short prompts and OpenWebText prompts with $L = 1$ is also shown. [Number of text outputs generated per parameter value $T$ and prompt: 1536. Number of generated output tokens: 10. Error bars indicate the standard error of the mean over all corresponding prompts. Error bars for the individual prompts in panel (b) are almost negligible and thus omitted to avoid visual clutter.]

The results for $L = 6$ are shown in Fig. 5(a) as colored lines, each corresponding to the distribution of the weights of a particular QKV layer. Different layers show transitions at roughly 20K (layer 5),

40K (layers 3), 50K (layer 4), and 80K (layer 4) epochs. We also observe a large peak around epoch 0, i.e., at the start of the training, highlighting that the LLM learns most rapidly at the beginning stages. In the long run, the dissimilarity curves approach 0, signaling that overall the weight distributions become less and less distinguishable.

Complementarily, in the same plot, we show dissimilarities derived from the LLM output distributions. The grey line corresponds to an average of the dissimilarities obtained by using entries from OpenWebText (Aaron Gokaslan and Vanya Cohen, 2019) (which serves as a proxy for the Pythia training dataset) as prompts. The black line corresponds to the average of results obtained from a selection of single-token prompts [see also panel (b)]. Both dissimilarity curves show a peak around epoch 0 as well as a peak around 80K epochs that is potentially related to the rapid change of layer 4 around the same time.

Figure 5(b) shows dissimilarities as a function of the training epoch for models queried with short, generic prompts ("", "0", "I", "?", "1", "You", and "!") at $L = 1$. These short prompts were selected to be as general as possible and the associated output distributions seem more sensitive as compared to the long examples from OpenWebText: their mean dissimilarity shows clear peaks near epochs 20K, 40K, and 80K. These correspond to outliers where the output distribution changes severely only at a single point and returns back (close) to its original behavior immediately after. As such, these peaks do not mark transitions between two macroscopic phases of behavior. We further verified this outlier behavior by inspecting the dissimilarity between the points directly to the left and right of the potential outlier. It remains an open question if these outliers are linked to the transitions observed in the layer weights shown in panel (a).

The larger $L$ value used in panel (a) averages out the signal stemming from these outliers. Such a reduced susceptibility to outliers can be an advantage of using $L \gg 1$ when searching for macroscopic transitions in particular.

Some peak locations in the dissimilarity curves are prompt-dependent, indicating that learning progresses differently for different types of behavior. Here we have used rather generic prompts, resulting in an analysis of the LLM's general behavior during training. However, in principle, conditioning on the prompt allows one to analyze whether and when specific knowledge emerges (Liu et al., 2021; Gurnee & Tegmark, 2023). As an outlook, one can imagine automatically monitoring changes across a multitude of prompts on different topics and testing different abilities at scale, without the need to design individual metrics for each prompt.

## 4 RELATED WORKS

Before concluding, let us discuss how our method relates to other approaches for studying phase transitions in physics and in LLMs.

**Generic performance-based analysis.** Many previous works found transitions in LLM behavior by locating sharp changes in generic performance measures, such as sudden drops in training loss (Olsson et al., 2022; Chen et al., 2023). While this may capture transitions in the overall behavior, such an approach cannot resolve transitions in specific LLM behavior. In particular, it may miss algorithmic transitions where the same performance is reached but by different means (Zhong et al., 2023).

**Prompt-specific success metrics.** Other works have found transitions by looking at success metrics tailored toward specific prompts (Austin et al., 2021; Brown et al., 2020; Hendrycks et al., 2020; Radford et al., 2021; Srivastava et al., 2022; Wei et al., 2022; Liu et al., 2021). Recalling the example studied in Sec. 3.1, this would correspond to assigning a score of 1 if the LLM provided the correct answer to the question $x < 42$ and 0 otherwise. Similarly, one could compute such a score in a temporal analysis (Sec. 3.3) or for detecting transitions as a function of another hyperparameter (Sec. 3.2). A downside of this approach is that it is restricted to prompts that allow for a clear score to be assigned. In particular, choosing an appropriate scoring function may require lots of human engineering. Moreover, discontinuous metrics can artificially induce transitions where the underlying behavior varies smoothly (Schaeffer et al., 2023) [3]. Similarly, they may miss transitions where the same performance is reached but by different means (Zhong et al., 2023).

---

[3] While spurious transitions can still occur with our method due to sampling noise, the metric we utilize is continuous and does not exhibit thresholding effects.

**Measures based on model internals.** The aforementioned approaches are based on the model output. Many works have also detected transitions based on changes in the internal structure of models, such as its trainable parameters (Millidge; Chen et al., 2023) (similar to the weight-based analysis we have performed in Sec. 3.3). However, access to model internals may not always be available. Moreover, the design of measures that capture specific transitions in behavior requires lots of human input (Räuker et al., 2023; Conmy et al., 2023; Zhong et al., 2023), e.g., using insights from the field of *mechanistic interpretability*.

## 5 CONCLUSION AND OUTLOOK

We have proposed a method for automating the detection of phase transitions in LLMs, and demonstrated that it successfully reveals a variety of transitions. Leveraging access to the LLMs' next-token probability distributions, the proposed dissimilarity measures can efficiently quantify distribution shifts without fine-tuning or adaption to the specific scenario at hand. Because the method is solely based on analyzing a model's output distribution and access to the model weights is not required, it enables *black-box interpretability* studies.

The proposed method is not only applicable to language models, but can be straightforwardly adapted to any generative model with an explicit, tractable density (Goodfellow, 2016; Arnold et al., 2023b). If one can draw samples from the output distribution but does not have explicit access to the underlying probabilities, then the dissimilarity measures can still be approximated using NN-based classifiers (Menon & Ong, 2016; Arnold et al., 2023a) tailored toward the particular data type, such as natural language.

**Limitations.** Future large-scale investigations are needed to fully understand how the uncovered transitions depend on variables such as the specific prompt, the number of generated output tokens, or the selected model. In particular, due to computational resource constraints, the size of the studied language models has been limited.

**Broader Impact.** Our method has the potential to enhance the development of future AI systems due to an improved understanding of their behavior. The dual-use nature of such systems carries inherent risks, which requires one to proceed with caution and implement mechanisms to ensure they are used safely and ethically.

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

## A    IMPLEMENTATION DETAILS

**Estimating dissimilarity measures.**    Our method for detecting phase transitions is based on estimating $D_g$ across the entire parameter range. Starting with a fixed set of points on a uniform grid $\mathcal{T}$, let us denote the set of in-between points at least $L$ points away from the border of the range as $\bar{\mathcal{T}}$ (note that $|\bar{\mathcal{T}}| = |\mathcal{T}| - 2L$). For each trial point $T^* \in \bar{\mathcal{T}}$, we obtain an unbiased estimate $\hat{D}_g = (\hat{J}_{\text{left}} + \hat{J}_{\text{right}})/2$, where

$$\hat{J}_i = \frac{1}{|\sigma_i|} \sum_{T \in \sigma_i} \frac{1}{\mathcal{D}(T)} \sum_{\boldsymbol{x} \in \mathcal{D}(T)} g \left[ P(\sigma_i | \boldsymbol{x}) \right]. \tag{A1}$$

Recall that $\sigma_{\text{left}}$ and $\sigma_{\text{right}}$ denote the $L$ closest points to the left or right of the trial point. Here, $\mathcal{D}(T)$ denotes a set of output texts $\boldsymbol{x}$ generated via the LLM at point $T \in \mathcal{T}$. In this work, we choose the number of generated text samples to be the same for all $T \in \mathcal{T}$, i.e., $|\mathcal{D}(T)| = |\mathcal{D}|$.

We perform the computation of $\hat{D}_g(T^*) \: \forall \, T^* \in \bar{\mathcal{T}}$ in two stages. In a first stage, we go through each grid point $T \in \mathcal{T}$ and generate text outputs that are $N_{\text{tokens}}$ in length via the LLM. The associated computation time scales as $|\mathcal{T}| \cdot N_{\text{tokens}} \cdot t_{\text{LLM,eval}}(|\mathcal{D}|)$, where $t_{\text{LLM,eval}}(|\mathcal{D}|) = \mathcal{O}(|\mathcal{D}|)$ corresponds to the time it takes the LLM to generate $|\mathcal{D}|$ different outputs (single token in length). In a second stage, for a given trial point $T^* \in \bar{\mathcal{T}}$, we evaluate the probability of each text output generated in its vicinity $\{ \boldsymbol{x} \in \mathcal{D}(T) | T \in \sigma_{\text{left}}(T^*) \cup \sigma_{\text{right}}(T^*) \}$ to come from the left or right segment. That is, we compute $P(\sigma_{\text{left}} | \boldsymbol{x})$ and $P(\sigma_{\text{right}} | \boldsymbol{x})$, i.e., a term in the sum of Eq. (A1). The computation time associated with the second stage scales as $|\bar{\mathcal{T}}| \cdot N_{\text{tokens}} \cdot t_{\text{LLM,eval}}(|\mathcal{D}|) \cdot 2L$. Note that in practice, one can embarrasingly parallelize over the different grid and trial points. Moreover, one generates and evaluates text outputs batchwise.

**Compute Resources.**    For our computations, we used an NVIDIA RTX 3090 GPU as well as NVIDIA Tesla V100 (32 GB) and NVIDIA A100 (40GB) GPUs.

**Code    availability.**    A    `Python`    implementation    of    our    method    is    available    at github.com/llmtransitions/llmtransitions.

**Utilized assets.**    Our code is implemented in `Python` and internally uses NumPy (Harris et al., 2020), PyTorch (Paszke et al., 2019), and transformers (Wolf et al., 2020). For the presented examples, we make use of the following additional packages: pandas (Wes McKinney, 2010), SciPy (Virtanen et al., 2020), Matplotlib (Hunter, 2007), and seaborn (Waskom, 2021).

The Pythia models, the Mistral models Mistral-7B-Instruct model and Mistral-7B-Base model, are available under the Apache-2.0 license on Hugging Face. The Meta-Llama-3-8B model and NVIDIA's Llama3-ChatQA-1.5-8B are available on Hugging Face under the Meta Llama 3 community license.

## B    THEORETICAL BACKGROUND ON $g$-DISSIMILARITIES

**Correspondence Between $f$-divergences and $g$-dissimilarities.**    Let us establish a correspondence between $f$-divergences [Eq. (1)] and $g$-dissimilarities [Eq. (4)]. In the following, we denote $P(\boldsymbol{x} | \text{left}, T^*)$ and $P(\boldsymbol{x} | \text{right}, T^*)$ as $P_{\text{left}}(\boldsymbol{x})$ and $P_{\text{right}}(\boldsymbol{x})$, respectively, and suppress the depen-

dence on $T^*$. We can write any $g$-dissimilarity as

$$D_g = \frac{1}{2} \left( \mathbb{E}_{\boldsymbol{x} \sim P_{\text{left}}} \left[ g\left[ P(\sigma_{\text{left}}|\boldsymbol{x}) \right] \right] + \mathbb{E}_{\boldsymbol{x} \sim P_{\text{right}}} \left[ g\left[ P(\sigma_{\text{right}}|\boldsymbol{x}) \right] \right] \right)$$

$$= \mathbb{E}_{\boldsymbol{x} \sim P_{\text{right}}} \left[ \frac{1}{2} \frac{P_{\text{left}}(\boldsymbol{x})}{P_{\text{right}}(\boldsymbol{x})} g\left[ P(\sigma_{\text{left}}|\boldsymbol{x}) \right] \right] + \mathbb{E}_{\boldsymbol{x} \sim P_{\text{right}}} \left[ \frac{1}{2} g\left[ P(\sigma_{\text{right}}|\boldsymbol{x}) \right] \right]$$

$$= \mathbb{E}_{\boldsymbol{x} \sim P_{\text{right}}} \left[ \frac{1}{2} \frac{P_{\text{left}}(\boldsymbol{x})}{P_{\text{right}}(\boldsymbol{x})} g\left[ P(\sigma_{\text{left}}|\boldsymbol{x}) \right] + \frac{1}{2} g\left[ P(\sigma_{\text{right}}|\boldsymbol{x}) \right] \right]$$

$$= \mathbb{E}_{\boldsymbol{x} \sim P_{\text{right}}} \left[ \frac{1}{2} \frac{P_{\text{left}}(\boldsymbol{x})}{P_{\text{right}}(\boldsymbol{x})} g\left[ \frac{P_{\text{left}}(\boldsymbol{x})}{P_{\text{left}}(\boldsymbol{x}) + P_{\text{right}}(\boldsymbol{x})} \right] + \frac{1}{2} g\left[ \frac{P_{\text{right}}(\boldsymbol{x})}{P_{\text{left}}(\boldsymbol{x}) + P_{\text{right}}(\boldsymbol{x})} \right] \right]$$

$$= \mathbb{E}_{\boldsymbol{x} \sim P_{\text{right}}} \left[ \frac{1}{2} \frac{P_{\text{left}}(\boldsymbol{x})}{P_{\text{right}}(\boldsymbol{x})} g\left[ \frac{\frac{P_{\text{left}}(\boldsymbol{x})}{P_{\text{right}}(\boldsymbol{x})}}{\frac{P_{\text{left}}(\boldsymbol{x})}{P_{\text{right}}(\boldsymbol{x})} + 1} \right] + \frac{1}{2} g\left[ \frac{1}{\frac{P_{\text{left}}(\boldsymbol{x})}{P_{\text{right}}(\boldsymbol{x})} + 1} \right] \right].$$

Thus, any $g$-dissimilarity $D_g$ can be rewritten in the form of an $f$-divergence $D_f[P_{\text{left}}, P_{\text{right}}]$ with

$$f(x) = \frac{x}{2} \cdot g\left( \frac{x}{1+x} \right) + \frac{1}{2} \cdot g\left( \frac{1}{1+x} \right). \tag{B1}$$

Note, however, that not any choice of $g$-function will lead to a proper $f$-divergence in the sense that the resulting $f$-function may not be convex and $f(1)$ may not be zero (recall the definition of an $f$-divergence in Sec. 2.1).

**JS divergence.** Using the correspondence above, we have that $D_{g(x) = \log(x) + \log(2)}$ is equivalent to an $f$-divergence $D_f[P_{\text{left}}, P_{\text{right}}]$ with

$$f(x) = \frac{x}{2} \cdot \log\left( \frac{2x}{1+x} \right) + \frac{1}{2} \cdot \log\left( \frac{2}{1+x} \right) \tag{B2}$$

which corresponds to the JS divergence.

**TV distance.** Let us further prove that the $D_{g(x) = 1 - 2\min\{x, 1-x\}}$ corresponds to the TV distance $D_{\text{TV}}[P_{\text{left}}, P_{\text{right}}]$. We have

$$D_{g(x) = 1 - 2\min\{x, 1-x\}} = 1 - \mathbb{E}_{\boldsymbol{x} \sim P_{\text{left}}} \left[ \min\{P(\sigma_{\text{left}}|\boldsymbol{x}), P(\sigma_{\text{right}}|\boldsymbol{x})\} \right]$$

$$- \mathbb{E}_{\boldsymbol{x} \sim P_{\text{right}}} \left[ \min\{P(\sigma_{\text{left}}|\boldsymbol{x}), P(\sigma_{\text{right}}|\boldsymbol{x})\} \right] \tag{B3}$$

Using the identity

$$\min\{P(\sigma_{\text{left}}|\boldsymbol{x}), P(\sigma_{\text{right}}|\boldsymbol{x})\} = \frac{1}{2} \left( 1 - |P(\sigma_{\text{left}}|\boldsymbol{x}) - P(\sigma_{\text{right}}|\boldsymbol{x})| \right),$$

Eq. (B3) can be rewritten as

$$D_{g(x) = 1 - 2\min\{x, 1-x\}} = \frac{1}{2} \mathbb{E}_{\boldsymbol{x} \sim P_{\text{left}}} \left[ |P(\sigma_{\text{left}}|\boldsymbol{x}) - P(\sigma_{\text{right}}|\boldsymbol{x})| \right]$$

$$+ \frac{1}{2} \mathbb{E}_{\boldsymbol{x} \sim P_{\text{right}}} \left[ |P(\sigma_{\text{left}}|\boldsymbol{x}) - P(\sigma_{\text{right}}|\boldsymbol{x})| \right]$$

$$= \frac{1}{2} \mathbb{E}_{\boldsymbol{x} \sim P_{\text{right}}} \left[ \left( 1 + \frac{P_{\text{left}}(\boldsymbol{x})}{P_{\text{right}}(\boldsymbol{x})} \right) |P(\sigma_{\text{left}}|\boldsymbol{x}) - P(\sigma_{\text{right}}|\boldsymbol{x})| \right]$$

$$= \frac{1}{2} \mathbb{E}_{\boldsymbol{x} \sim P_{\text{right}}} \left[ P(\sigma_{\text{right}}|\boldsymbol{x}) \left( 1 + \frac{P_{\text{left}}(\boldsymbol{x})}{P_{\text{right}}(\boldsymbol{x})} \right) \left| 1 - \frac{P(\sigma_{\text{left}}|\boldsymbol{x})}{P(\sigma_{\text{right}}|\boldsymbol{x})} \right| \right].$$

Noting that $\frac{P(\sigma_{\text{left}}|\boldsymbol{x})}{P(\sigma_{\text{right}}|\boldsymbol{x})} = \frac{P_{\text{left}}(\boldsymbol{x})}{P_{\text{right}}(\boldsymbol{x})}$ and $P(\sigma_{\text{left}}|\boldsymbol{x}) + P(\sigma_{\text{right}}|\boldsymbol{x}) = 1$, we finally obtain

$$D_{g(x) = 1 - 2\min\{x, 1-x\}} = \mathbb{E}_{\boldsymbol{x} \sim P_{\text{right}}} \left[ \frac{1}{2} \left| 1 - \frac{P_{\text{left}}(\boldsymbol{x})}{P_{\text{right}}(\boldsymbol{x})} \right| \right]$$

which corresponds to an $f$-divergence $D_f[P_{\text{left}}, P_{\text{right}}]$ with $f(x) = \frac{1}{2}|1 - x|$, i.e., the TV distance.

**Freedom in Choice of $g$-function.** Note that the choice of $g$-function leading to a particular $g$-dissimilarity is not unique. In particular, we have that $D_{\tilde{g}} = D_g$ for any $\tilde{g}(x) = g(x) + c(\frac{1}{x} - 2)$ where $c \in \mathbb{R}$ is some constant:

$$D_{\tilde{g}} = D_g + \frac{c}{2}\left( \mathbb{E}_{\boldsymbol{x} \sim P_{\text{left}}}\left[ \frac{1 - P(\sigma_{\text{left}}|\boldsymbol{x})}{P(\sigma_{\text{left}}|\boldsymbol{x})} - 1 \right] + \mathbb{E}_{\boldsymbol{x} \sim P_{\text{right}}}\left[ \frac{1 - P(\sigma_{\text{right}}|\boldsymbol{x})}{P(\sigma_{\text{right}}|\boldsymbol{x})} - 1 \right] \right)$$

$$= D_g + \frac{c}{2}\left( \mathbb{E}_{\boldsymbol{x} \sim P_{\text{left}}}\left[ \frac{P(\sigma_{\text{right}}|\boldsymbol{x})}{P(\sigma_{\text{left}}|\boldsymbol{x})} - 1 \right] + \mathbb{E}_{\boldsymbol{x} \sim P_{\text{right}}}\left[ \frac{P(\sigma_{\text{left}}|\boldsymbol{x})}{P(\sigma_{\text{right}}|\boldsymbol{x})} - 1 \right] \right)$$

$$= D_g + \frac{c}{2}\left( \mathbb{E}_{\boldsymbol{x} \sim P_{\text{left}}}[1] + \mathbb{E}_{\boldsymbol{x} \sim P_{\text{right}}}[1] - 2 \right) = D_g.$$

**Relation to the Fisher information.** In the following, we prove that any $g$-dissimilarity with $g(\frac{1}{2}) = 0$ and a twice-differentiable $g$-function reduces to the Fisher information in lowest order.

For this, consider the case with $L = 1$ where we compare the distributions at two points in parameter space that are separated by $\delta T$. The corresponding $g$-dissimilarity is equivalent to an $f$-divergence $D_f[P(\boldsymbol{x}|T), P(\boldsymbol{x}|T + \delta T)]$. Note that $D_f[P(\boldsymbol{x}|T), P(\boldsymbol{x}|T)] = 0$ if $f(1) = 0$ and $\partial D_f[P(\boldsymbol{x}|T), P(\boldsymbol{x}|B)]/\partial B|_{B=T} = f'(1)\partial \mathbb{E}_{\boldsymbol{x} \sim P(\cdot|T)}[1]/\partial T = f'(1)\partial 1/\partial T = 0$. The second-order derivative corresponds to

$$\left. \frac{\partial^2 D_f[P(\boldsymbol{x}|T), P(\boldsymbol{x}|B)]}{\partial B^2} \right|_{B=T} = f'(1)\mathbb{E}_{\boldsymbol{x} \sim P(\cdot|T)}\left[ \frac{1}{P(\boldsymbol{x}|T)} \frac{\partial^2 P(\boldsymbol{x}|T)}{\partial T^2} \right]$$

$$+ f''(1)\mathbb{E}_{\boldsymbol{x} \sim P(\cdot|T)}\left[ \left( \frac{\partial \log P(\boldsymbol{x}|T)}{\partial T} \right)^2 \right]$$

$$= f''(1)\mathcal{F}(T)$$

assuming $f'(1) = 0$, where $\mathcal{F}$ is the Fisher information. Thus, for $L = 1$ and parameter values separated by $\delta T$, we can express any $g$-dissimilarity with $g(\frac{1}{2}) = 0$ as $D_g = \frac{g''(\frac{1}{2})}{32}\mathcal{F}(T)\delta T^2 + \mathcal{O}(\delta T^3)$. The fact that $f'(1)$ must be zero translates into the condition that $g'(\frac{1}{2}) = 0$. This is not a fundamental restriction since we have some freedom in the choice of $g$ function. That is, we can replace $g \mapsto \tilde{g}$, where $\tilde{g}(x) = g(x) + c(\frac{1}{x} - 2)$ with $c = \frac{1}{6}g'(\frac{1}{2})$, retaining $D_g = D_{\tilde{g}}$ and ensuring that $\tilde{g}'(\frac{1}{2}) = 0$.

# C   DETAILS ON ENERGY-BASED ANALYSIS OF TEMPERATURE TRANSITION

Let $\boldsymbol{x} = (x_1, \ldots, x_N)$ be a sequence of $N$ tokens generated for a fixed prompt from an autoregressive LLM such as the ones considered in this article. The distribution of $\boldsymbol{x}$ is given by

$$P(\boldsymbol{x}|T) = Q_T(x_N|x_1, \ldots, x_{N-1})Q_T(x_{N-1}|x_1, \ldots, x_{N-2}) \cdots Q_T(x_2|x_1)Q_T(x_1),$$

denoting the fact that the tokens are sampled sequentially. In each step, a token is sampled from a Boltzmann distribution $Q_T$,

$$Q_T(x_i|x_1, \ldots, x_{i-1}) = e^{-E(x_i|x_1, \ldots, x_{i-1})/T}/Z_i(T).$$

Here, $Z_i(T) = \sum_{x_i} e^{-E(x_i|x_1, \ldots, x_{i-1})/T}$ is a normalization factor with the sum running over all possible $i$th tokens. The conditional energies, $E(x_i|x_1, \ldots, x_{i-1})$ are typically referred to as logits and learned from data. Note that while the distribution over individual tokens can be expressed as a Boltzmann distribution at varying temperature, the overall distribution $P(\boldsymbol{x}|T)$ cannot.[4]

Nevertheless, we can define an energy scale for the entire system by viewing the overall probability distribution at $T = 1$ as a Boltzmann

$$P(\boldsymbol{x}|T = 1) = e^{-E(\boldsymbol{x})}/Z, \tag{C1}$$

where $Z$ is a normalization constant independent of $\boldsymbol{x}$ (also referred to as partition function). Recall that any valid probability distribution can be written in the form of Eq. (C1) with a suitably chosen energy function. Taking the logarithm of Eq. (C2) and reordering, we have

$$E(\boldsymbol{x}) = -\log P(\boldsymbol{x}|T = 1) - \log Z. \tag{C2}$$

---

[4]In order for a quantity to be a valid energy of a system, it cannot itself depend on temperature, i.e., change with temperature.

Using Eq. (C2), we can compute the total energy up to the constant $-\log Z$ which serves as our reference point for the energy scale. In the main text, we use $-\log P(\boldsymbol{x}|T=1)$ as the total energy to compute and compare energy statistics at various temperatures. In particular, the heat capacity remains the same under the mapping $-\log P(\boldsymbol{x}|T=1) - \log Z \mapsto -\log P(\boldsymbol{x}|T=1)$.

# D PHYSICS RELATED TERMINOLOGY

## D.1 PHASE TRANSITIONS IN PHYSICS

The most common definition of a phase transition in physics has been given through the Ehrenfest classification, which demands that the $n$th derivative of the free energy must have a discontinuity in the thermodynamic limit of an infinitely large system size. There are, however, notable exceptions to this definition, such as the Kosterlitz-Thouless transition, which is of infinite order and involves topological order. Moreover, this classification requires the identification of free energy and is thus restricted to the context of equilibrium systems.

Besides phase transitions, there are also less stringent, related phenomena like crossovers, which do not require strict discontinuities. As such, even in physics, the definitions of phase transitions are being debated and have to be reevaluated when venturing into novel domains.

In the context of LLMs, the term "phase transition" is often used rather colloquially to draw an analogy to the physical phenomenon. For the present article, we have used the loose but pragmatic definition of a phase transition referring to a rapid change of the system's state as the tuning parameter is varied.

## D.2 HEAT CAPACITY

The heat capacity is the derivative of a system's mean energy with respect to its temperature. In non-physics terms, the mean energy can be understood as being proportional to the mean logarithmic probability of the system's state. Because it can be related to derivatives of the free energy, or similarly, to derivatives of the underlying probability distribution, divergences in this quantity are signatures of phase transitions. In a physical equilibrium setting, it is also closely related to the Fisher information.

## D.3 ISING AND POTTS MODELS

A very prominent example of a phase transition relevant to the present work is the Ising model phase transition in one dimension. The Ising model is better known as a Boltzmann machine or Hopfield network in the machine learning community and described by the energy $E = -\sum_{ij} w_{ij} s_i s_j$ in its simplest form. If the variables $s$ have more than two possible states, this becomes a Potts model.

In one dimension and as a function of temperature, such systems can undergo phase transitions in the Ehrenfest sense involving a discontinuity only if the range of the interaction matrix $w$ is infinite. A language model's output can be viewed as a one-dimensional system of tokens whose interaction with each other is mediated by the attention mechanism. In particular, the correspondence between self-attention and Potts models has recently been formalized in (Rende et al., 2024).

Note that the interaction only occurs in the forward direction for auto-regressive models and is of finite range due to the finite context length of current models. While this cannot give rise to a strict discontinuity, rapid changes qualify for our weaker definition of a phase transition (or a crossover), and by scaling the system size appropriately one may observe similar critical behavior as in physics.

# E   ADDITIONAL SCANS FOR VARYING PROMPT

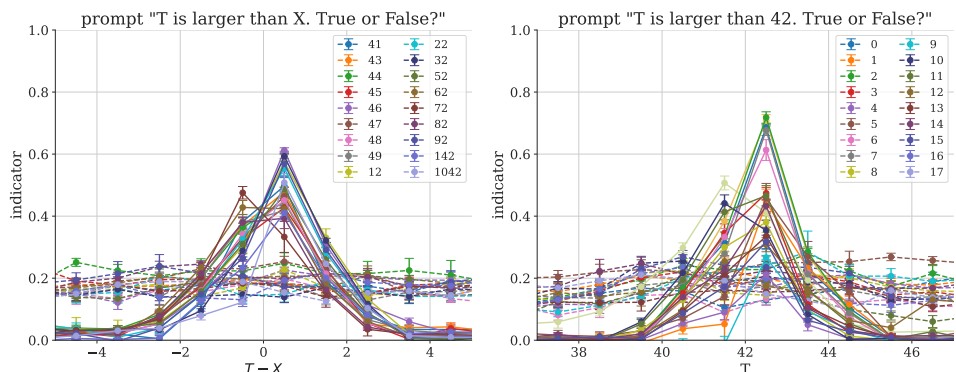

Figure E1: Additional results for integer scan of Fig. 1 using the same numerical settings with the Mistral-7B-Instruct model (full lines) and the Mistral-7B-base model (dashed lines). (Left) Linear dissimilarity as a function of $T - X$, where $X$ takes on a representative range of values (see legend). (Right) Linear dissimilarity as a function of $T$ for various rephrasings of the original prompt "T is larger than 42. True or False?" obtained from ChatGPT4 (numbering in the legend refers to different prompt variations). Examples include "Is T greater than 42?" and "Would you consider T to be more than 42?". In both plots, the peak for the instruct models is always at either $X - 0.5$ or $X + 0.5$, as expected (where $X = 42$ for the right plot).

# F   ADDITIONAL SCANS FOR VARYING TEMPERATURE

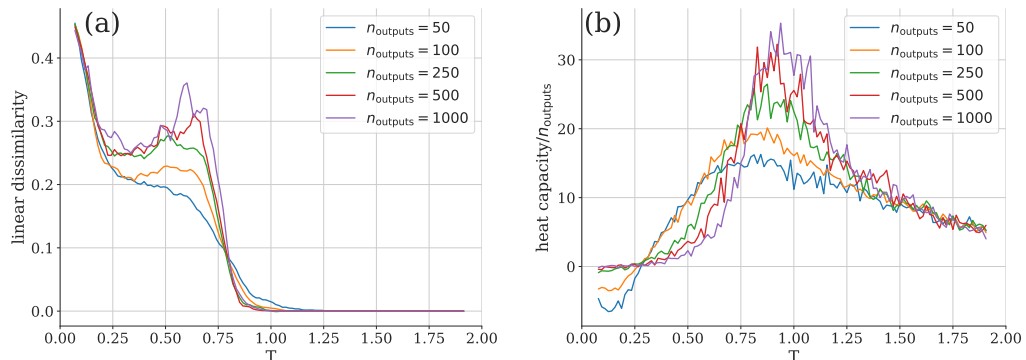

Figure F1: High-temperature transition of Pythia 70M model in response to the prompt "The opinions expressed by columnists are their own and do not represent the views of Townhall.com.\n\n" – an excerpt from OpenWebText (Aaron Gokaslan and Vanya Cohen, 2019). (a) Linear dissimilarity measure ($L = 5$) and (b) heat capacity for various numbers of generated output tokens $n_{\text{outputs}}$ with the temperature range $[10^{-4}, 2]$. [Number of text outputs generated per parameter value $T$: 1000, 1000, 300, 100, and 100 for $n_{\text{outputs}} = 50, 100, 250, 500, \text{and } 1000$, respectively.]

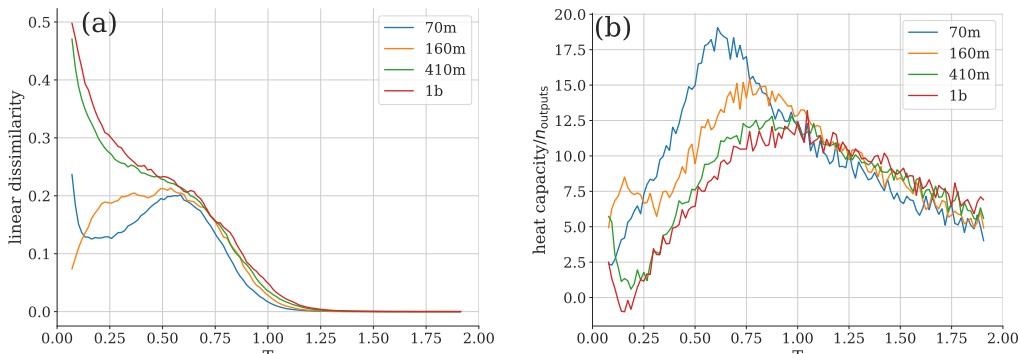

Figure F2: High-temperature transition of for Pythia models of various sizes in response to the prompt "There's measuring the drapes, and then there's measuring the drapes on a house you haven't bought, a" – an excerpt from OpenWebText (Aaron Gokaslan and Vanya Cohen, 2019). (a) Linear dissimilarity measure ($L = 5$) and (b) heat capacity with the temperature range $[10^{-4}, 2]$. [Number of text outputs generated per parameter value $T$: 1000. Number of generated output tokens $n_{\mathrm{outputs}}$: 50.]

