# OpenReview forum: "Phase Transitions in the Output Distribution of Large Language Models"
_ICLR.cc/2025/Conference — Submitted to ICLR 2025_

### Official Review · Reviewer_4pP5 · 2024-11-02

**Soundness:** 2
**Presentation:** 3
**Contribution:** 2
**Rating:** 5
**Confidence:** 3

**Summary:**

This paper examines phase transition in the output distributions of large language models(LLMs) and proposes a novel approach that applies a method used in physical systems for detecting phase transitions to LLMs. In physical systems, phase transitions occur as control parameters, such as temperature and pressure, change. These transitions often depend only on macroscopic parameters and show universal properties. However, phase transitions in LLMs have been shown to depend on detailed ``microscopic" inputs, such as prompts, with phase transitions occurring at different temperatures depending on the prompt. This difference suggests that a distinct mechanism may be at work in LLMs compared to physical systems, and the fact that the "phase of behavior" in LLMs is induced by prompts is particularly interesting. This study provides a new method for automatically detecting the phase transitions in LLMs through black-box analysis based on output distribution, yielding new insights into the learning behaviors of LLMs.

**Strengths:**

* This study is significant for applying a phase transition detection method from physics to the output distribution of LLMs. In physics, order parameters are often known, making the method less critical. However, in LLMs, the order parameter is entirely unknown, allowing this method to show its full advantages.

* Considering the complex internal structure of LLMs, often regarded as a black box, this approach provides a potentially valuable tool for understanding LLM properties. As the authors point out, this is an advantage as it can be implemented even without detailed knowledge of an LLM's internal structure.

**Weaknesses:**

* Although multiple phase transitions are identified, it remains unclear how they are distinguished from crossover, as mentioned in Appendix. For example, the study suggests three phases when temperature is changed, but there is a lack of explanation of what each phase represents and its significance from a linguistic perspective.

* This point slightly contradicts the above strength: what does it mean for phase transitions to depend on prompts? For example, if the temperature at which water freezes changes depending on the shape of the glass, can we truly say water undergoes a phase transition? While I understand the significance of the authors' extensive numerical experiments, it is difficult to understand their relevance to LLM research.

**Questions:**

*I would like clarification on how the authors calculated the heat capacity. In connection with this, it would be valuable if the authors could provide concrete evidence or specific analyses regarding the occurrence of negative heat capacity. In typical thermodynamic systems, heat capacity should be positive; however, systems with long-range interactions or non-equilibrium properties can exhibit negative values. Have the authors investigated such properties in LLMs, for instance, through analyses of long-range interactions such as the attention mechanism, or do they have evidence supporting this phenomenon in LLMs?
* The advantage of the method used by the authors is that it can detect phase transitions from the output distributions. However, what kind of phase can the authors say it represents? Additionally, can the authors specify the type of phase transition, such as whether it is first-order or second-order?
*Minor points.
** It would be better to write the number "-1" in line 166 in math mode.
** The caption for Fig.1 seems incorrect regarding the reference to the Appendix.
** I initially thought Fig.4(b) was an enlarged view of Fig.4(a). While specific heat naturally connects, linear dissimilarity appears not to connect. Could the authors explain the reason?

---

### Official Review · Reviewer_mqhi · 2024-11-03

**Soundness:** 3
**Presentation:** 3
**Contribution:** 2
**Rating:** 3
**Confidence:** 4

**Summary:**

This paper looks at LLMs from the perspective of statistical physics. It proposes several ways of measuring changes in probability distributions, where large changes are interpreted as signatures for "phase transitions". It applies these measures in 3 case studies. In the first, change is measured as a simple prompt, containing a numerical value, is varied. In the second, the temperature parameter in the softmax is varied. In the third, changes across checkpoints during training are recorded. For each of these case studies, graphs are produced, showing that there are ranges for the varied parameter where changes are larger.

**Strengths:**

This paper addresses an interesting problem: whether changes in output distributions of LLMs, as one varies an underlying parameter, can be used to detect interesting changes in the underlying mechanisms. It reviews some potentially useful measures from statistical physics, that place widely used measures in NLP and ML such as the KL divergence in a broader context.

The authors have looked at many different models and checkpoints, and obtaining the numbers reported in this paper / creating the graphs must have been quite a lot of work.

**Weaknesses:**

The first half of the paper is devoted to reviewing mathematical tools to measure distances between distributions. In the second half of the paper, the distance measures are used in some simple case studies. The main weakness of the paper is that each of these case studies stops where it starts to become interesting, such that we learn nothing new about the behaviour or underlying mechanisms of LLMs, and that the usefulness of the mathematical tools from the first half is never shown.

I don't have a lot of comments on the distance measures from the first 3-4 pages. Countless papers in ML and NLP already measure distances between probability distributions (usually with KL and computationally tractable approximations such as ELBO). It would be surprising and potentially important if tools developed for studying phase transitions in physics offer a new light on these issues. I cannot judge how novel the framework the authors offer is for ML, but novel or not, I will just note that it is problematic that the authors do not in any way show that their favourite tools are in fact more useful than better known tools.

This usefulness would have to be established in the case studies in the second half, but unfortunately isn't. In case study (1), figure 1 and 2 reveals that LLMs build up internal representations for the numerical value of integers, that some do worse than others and that tokenization sometimes messes up the representations. None of that is news for people that have studied LLMs, and especially for people that have read papers on numerosity representations. And, moreover, it has nothing to do with phase transitions, unless that term is given such a broad definition that it looses its value. The fact that output distributions change significantly around the point where the input necessitates a different response is simply a consequence of designing and training networks to predict categorical output.

In case study (2), we get some graphs of how sensitive output distributions are for different values of the temperature parameter in the output softmax. The graphs look pretty, but what have we learned? My conclusions from reading this section is that for high temperature, distributions do not change (because they're random anyway), and that for low temperatures the exact value might have a quite big impact on the behaviour of the LLM (but that we don't know beforehand at which value and which effect). But all of that is pretty much the starting point of the section.

In case study (3), the paper studies training dynamics in the Pythia model suite. Figure 5 looks interesting, especially the different timing of transitions in layer 0-2, 3, 4 and 5 (although it is not clear to me which Pythia model this is exactly, and why are you only plotting layer 0-5). But then the text (line 449) explains that some of the peaks (in 5b at least) might just be noise, and not reflecting any macroscopic phase changes. So again, we are presented some graphs that may or may not show that some interesting changes in the underlying representations might be happening, here at specific moments during training. But the paper presents no effort to actually confirm that these changes are indeed "phase transitions" with a different method, or in any way characterize the representation before and after the "phase transition".

All in all, there is clearly a lot of effort and technical know-how that has gone into this project, but it suffers from a lack of engagement with the extensive literature on LLMs, their learned representations and probability distributions, the qualitative changes in these representations, and the potential usefulness of concepts like "phase transitions" and "emergence" in understanding those changes. In section 4, the authors cite a lot of relevant papers (e.g., Wei et al 22, Schaeffer et al. 24, Chen et al. 23), but without actually engaging with them. For this project to really have some impact on ML/NLP, more is required than plotting the values of a measure for lots of different models and lots of different parameters: you need to show that high values on these measures indeed reveal something important, by showing this importance with a different method yourself or by linking it to a result from an existing paper.

EDIT: I have read the general response and the responses to my and other reviews. Although the direct response to my review was brief, overall I appreciate that the authors have put quite a bit of work into answering questions and concerns from the reviewers. It's clear that the authors are knowledgable about work of phase transitions in physics; however, their responses don't take away my concern that the main work -- to demonstrate the usefulness of porting this framework to ML -- still needs to be done. I will therefore maintain my score (as well as my recommendation for the future to engage with some existing work in ML, and investigate whether new tools can shed new light on it!).

**Questions:**

Questions:
Which of the Pythia models is used for the graphs in figure 5? Why are you only plotting layer 0-5?

EDIT: Thanks for your answer.

Suggestions:

Perhaps interesting work to engage with related to case study 1: Hanna, M., Liu, O., & Variengien, A. (2024). How does GPT-2 compute greater-than?: Interpreting mathematical abilities in a pre-trained language model. Advances in Neural Information Processing Systems, 36. This paper reveals a circuitry involved in building up a numerical prediction; the emergence of such a circuitry during training, or the switching on or off of such a circuitry due to contextual cues (in-context learning) would be an actually interesting phase transition.

Related to case study 3: Nanda, N., Chan, L., Lieberum, T., Smith, J., & Steinhardt, J. (2023). Progress measures for grokking via mechanistic interpretability. arXiv preprint arXiv:2301.05217. This paper shows you can predict the *sudden* emergence of a specific numerical ability (a phase transition) by tracking the *gradual* evaluation of specific "progress measures".

---

### Official Review · Reviewer_7XoD · 2024-11-03

**Soundness:** 2
**Presentation:** 2
**Contribution:** 2
**Rating:** 3
**Confidence:** 4

**Summary:**

The authors propose statistical methods to automatically detect phase transitions in large language models. They provide empirical evidence for three types of phase transitions that occur when varying the input, temperature, and training epoch.

**Strengths:**

The connection between phase transitions in physics and large language models is interesting!

**Weaknesses:**

The paper's main challenge and contribution are unclear. While the authors cite several previous works that have analyzed phase transitions, their contribution appears to be proposing new statistical detection methods. However, if this is their primary contribution, the experimental validation is insufficient - they use only a limited number of prompts and test cases. More extensive experiments with diverse prompts would be needed to validate their methods.

Alternatively, if the paper aims to establish theoretical connections to physics, it requires more accessible and thorough explanations. For example, while they discuss the link between the Ising model and language models, readers unfamiliar with physics may struggle to follow the comparison without mathematical formalization. Also, the paper would be more effective if it included parallel visualizations comparing phase transitions in physical systems (e.g., heat capacity plots for water) with their language model findings.

**Questions:**

1. What advantages does linear dissimilarity offer compared to total variation distance and Jensen-Shannon divergence? Figure 1 shows that all types of divergence can capture the dissimilarities.
2. It would be interesting to examine how phase transitions occur given combinations of input and temperature, similar to how water's phase transitions depend on both pressure and temperature.
3. What are the different properties of phases before and after transition? Can these phases be analogous to physical states like 'solid' or 'gas'? Is there a human-interpretable explanation for each phase, particularly regarding temperature?
4. What is the definition of a phase transition? In Figure 2, the linear dissimilarity of 0.4 for the Llama3 base model appears quite high. Can this be considered a phase transition?
5. Is panel (b) in Figure 4 truly a zoomed-in plot of panel (a)? The black line and y-axis values appear to be different.
6. In Figure 5(b), why did you mention that the clear peaks near epochs 20K, 40K, and 80K are outliers?"

---

### Official Review · Reviewer_aK6V · 2024-11-04

**Soundness:** 2
**Presentation:** 3
**Contribution:** 2
**Rating:** 3
**Confidence:** 3

**Summary:**

This paper investigates phase transitions in the output distributions of language models – that is, situations where the output distribution of an LM abruptly changes when a numerical parameter crosses some value. Changes are measured using f-divergences, where f is a specific rational function. The paper investigates three example parameters: (i) a number appearing in the context [concretely, a number “T” as in “T is larger than 42”], (ii) the softmax temperature in the output, (iii) the number of training epochs.

**Strengths:**

- The paper is innovative, linking physics concepts with language models
- The paper studies multiple open LLMs
- The proposed metric is theoretically grounded in a link to the Fisher information
- I found the execution to be rigorous and writing to be clear
- Findings can be of potential interest, e.g. it might be surprising that there phase transitions are identified at some training epochs

**Weaknesses:**

From my perspective, the primary weakness of the paper is the lack of any external validation of the proposed approach. While the paper proposes a method for measuring phase transitions, it remained unclear to me how to validate the findings. The findings might in principle depend a lot on the choice of the divergence. Is there any independent way of validating that the outcomes of the proposed method are useful or interesting? Is there a clear use case? Is there a principled relation to a theoretical property of language, or of transformer networks? Without this, it is not clear to me how to (i) substantiate the claim that the proposed metric captures transitions in an interesting way, (ii) make the paper of interest to the community. I do believe that such things are possible, and that the proposed method can be of substantial interest, but it would take further work to do this, and to validate the proposed approach specifically, compared to various other ways one could measure phase transitions (e.g., with other f-divergences)

**Questions:**

See Weaknesses: How do we know the method captures something interesting? Is there external validation?

---

### Meta-Review · Area_Chair_zhDJ · 2024-12-20

**Metareview:**

This paper applies mathematical ideas about phase transitions to study language models. Though reviewers found this interesting in the abstract, there was a general confusion about what question is actually being addressed here, and what these methods really bring to the table. It seems that more clarity is required before this paper is ready for publication. See in particular the discussion from reviewer mqhi

**Additional Comments On Reviewer Discussion:**

see above

---

### Decision · Program_Chairs · 2025-01-22

Reject